# MINIMALIST AND HIGH-PERFORMANCE SEMANTIC SEGMENTATION WITH PLAIN VISION TRANSFORMERS

## ABSTRACT

In the wake of Masked Image Modeling (MIM), a diverse range of plain, non-hierarchical Vision Transformer (ViT) models have been pre-trained with extensive datasets, offering new paradigms and significant potential for semantic segmentation. Current state-of-the-art systems incorporate numerous inductive biases and employ cumbersome decoders. Building upon the original motivations of plain ViTs, which are simplicity and generality, we explore high-performance 'minimalist' systems to this end. Our primary purpose is to provide simple and efficient baselines for practical semantic segmentation with plain ViTs. Specifically, we first explore the feasibility and methodology for achieving high-performance semantic segmentation using the last feature map. As a result, we introduce the PlainSeg, a model comprising only three 3×3 convolutions in addition to the transformer layers (either encoder or decoder). In this process, we offer insights into two underlying principles: (i) high-resolution features are crucial to high performance in spite of employing simple up-sampling techniques and (ii) the slim transformer decoder requires a much larger learning rate than the wide transformer decoder. On this basis, we further present the PlainSeg-Hier, which allows for the utilization of hierarchical features. Extensive experiments on four popular benchmarks demonstrate the high performance and efficiency of our methods. They can also serve as powerful tools for assessing the transfer ability of base models in semantic segmentation. The codes will be available.

## 1 INTRODUCTION

In modern semantic segmentation models, a typical architecture comprises a hierarchical pre-trained backbone serving as the feature extractor and a decoder adapting the extracted features to per-pixel predictions Long et al. (2015); Ronneberger et al. (2015); Badrinarayanan et al. (2017). In the CNN era, ResNets He et al. (2016) with different depths are widely adopted as feature extractors. On top of the hierarchical features extracted by backbones, various decoders are proposed to extract contextual information and refine feature maps Ghiasi & Fowlkes (2016); Chen et al. (2017a); Zhao et al. (2017); Lin et al. (2017a); Peng et al. (2017); Zhang et al. (2018). For the last two years, vision transformers were introduced to semantic segmentation and showed significant improvements over strong CNN models Zheng et al. (2021); Liu et al. (2021); Xie et al. (2021); Strudel et al. (2021); Yuan et al. (2021b). Naturally, researchers have explored decoder designs tailored for ViT backbones Zheng et al. (2021); Xie et al. (2021); Cao et al. (2021); Yan et al. (2022). A promising discovery is that cumbersome decoders are unnecessary for ViT-based models Xie et al. (2021), which brings out a series of practical segmentation models relying on efficient backbones and decoders Xie et al. (2021); Gu et al. (2022); Guo et al. (2022). However, most of the existing works in this area have focused on hierarchical ViTs, which are considered more suitable for dense prediction tasks than plain ViTs.

Recently, masked image modeling (MIM) has given rise to potent and scalable pre-trained plain ViTs, which substantially outperform their supervised counterparts in downstream tasks He et al. (2022); Bao et al. (2021); Zhou et al. (2021); Peng et al. (2022). The application of plain ViTs, originally designed for natural language processing (NLP), to the domain of semantic segmentation, is an intriguing endeavor as it contributes to the development of a more versatile foundational model capable of excelling in two distinct tasks. Early works such as SETR Zheng et al. (2021) and DPT Ranftl et al. (2021) design simple convolution-based decoders to adapt plain ViTs for semantic segmentation. In Segmenter Strudel et al. (2021), the authors propose a mask transformer

to construct a full transformer model for semantic segmentation. More recently, SegViT Zhang et al. (2022) proposes the attention-to-mask (ATM) module to harness the off-the-shelf features of plain ViTs. In contrast, ViT-Adapter Chen et al. (2022b) introduces an auxiliary top-down branch with numerous induction biases to unleash the potential of plain backbones. Notably, ViT-adapter with the Mask2Former Cheng et al. (2022) framework achieves state-of-the-art performance on multiple semantic segmentation datasets. However, despite its very impressive performance and considerable influence, we note that there are several overlooked issues with this approach. Firstly, when viewed within the context of the complete system, the introduction of numerous induction biases contradicts the original essence of plain vision transformers. Subsequent works built upon it may result in increasingly complex systems. If the sole objective is achieving state-of-the-art performance, hierarchical backbones Wang et al. (2022a) may offer a more suitable alternative. Secondly, from the perspective of transfer learning, employing cumbersome decoders is unnecessary. This is because a larger number of randomly initialized parameters typically necessitates a greater amount of training data and labeled samples from downstream tasks to realize their full potential, which deviates from the primary goal of transfer learning. There is a fact that the proportion of randomly initialized parameters to pre-trained parameters is close to 90 % for the state-of-the-art ViT-L-Adapter-Mask2Former.

In the light of the above analysis, we aim to develop high-performance 'minimalist' systems for segmentation segmentation with plain ViTs. The 'minimalist' pursuit encompasses both the reduction of inductive biases and the simplification of decoders, which are inherently interconnected. Recent research in objective detection Li et al. (2022) and human pose estimation Xu et al. (2022) has suggested that leveraging the last feature maps from MIM pre-trained plain ViTs, along with simple decoders, can yield satisfactory results. It implies that plain backbones can learn the prior knowledge from data, rendering the sophisticated and dedicated designs unnecessary. These findings establish the foundation for the feasibility of high-performance 'minimalist' systems in semantic segmentation. Motivated by these insights, we initially opt for a hard setting, focusing solely on utilizing the output of the last transformer layer (eliminating the use of hierarchical features). The proposed method named **PlainSeg** consists of a pre-trained plain ViT, three $3\times3$ convolution layers, and several lightweight transformer decoder layers, as illustrated in Fig. 1. While developing such systems, we offer insights into the underlying principles that render efficacy and promote efficiency: (i) high-resolution features are crucial to high performance in spite of employing simple up-sampling techniques and (ii) the slim transformer decoder requires a much larger learning rate than the wide transformer decoder, for instance, $10\times$ difference. On this basis, we further present **PlainSeg-Hier** which incorporates hierarchical features, as seen in previous works. We benchmark the proposed methods on four popular datasets (*i.e.* ADE20K, PASCAL Context, COCO-Stuff-10K, and COCO-Stuff-164K) with various pre-trained plain ViTs, reporting performance and inference efficiency of numerous models.

Our contributions can be summarized in the following three aspects: (i) we develop high-performance 'minimalist' systems for semantic segmentation with plain ViTs, which achieve highly competitive performance compared to ViT-Adapter and outperform SegViT; (ii) we offer insights into the practical principles for adapting potent plain ViTs to semantic segmentation tasks; (iii) the combination of high performance, elegant simplicity, and efficient inference and parameter utilization in our methods establishes solid baselines for future research in this field. Moreover, these methods serve as powerful tools for assessing the transfer ability of forthcoming plain ViT backbones in the context of semantic segmentation.

Note that we do not claim any algorithmic superiority over the current state-of-the-art. We do not conduct complete and fair comparison experiments with previous methods. This is difficult due to the huge training cost and not our intent. As stated, the contributions of this study are simple and efficient baselines and several practical principles.

## 2 RELATED WORK

**Vision Transformers.** Since the introduction of transformer architecture in NLP by Dosovitskiy et al. Vaswani et al. (2017); Dosovitskiy et al. (2020), vision transformers have made measurable progress in the field of computer vision. There are two main streams during the development of vision transformers: some introduce the inductive biases of convolutional networks such as the hierarchical features and locality of convolutions for better image modeling Wang et al. (2021); Liu et al. (2021);

Yuan et al. (2021a); Chu et al. (2021); Wu et al. (2021); Wang et al. (2022b) while others explore the potential of original (plain) transformers. The latter covers the improvement of supervised training strategies Touvron et al. (2021); Steiner et al. (2021); Touvron et al. (2022), masked image modeling for self-supervised learning He et al. (2022); Bao et al. (2021); Zhou et al. (2021); Chen et al. (2022a); Peng et al. (2022); Fang et al. (2022), and the utilization of multi-model data Wang et al. (2022c); Fang et al. (2022). In this study, we transfer pre-trained plain ViTs to semantic segmentation and continue its 'minimalist' pursuit.

**General Semantic Segmentation.** There have been numerous works in applying convolution neural networks (CNN) to semantic segmentation Chen et al. (2017a); Zhao et al. (2017); Peng et al. (2017); Lin et al. (2017a); Zhang et al. (2018); Chen et al. (2017b); Fu et al. (2019); Huang et al. (2019); Takikawa et al. (2019); Yuan et al. (2021c). They generally follow the encoder-decoder paradigm established by seminal works Long et al. (2015); Ronneberger et al. (2015). Due to the limited receptive fields of local convolutions, previous works mainly aim at better capturing contextual information, in both encoders and decoders Chen et al. (2017a); Zhao et al. (2017); Peng et al. (2017); Zhang et al. (2018); Chen et al. (2017b); Fu et al. (2019); Huang et al. (2019); Yuan et al. (2021c). With the rise of vision transformers, recent research shows that the performance can be boosted by only replacing the CNN backbones with various pyramid ViTs Liu et al. (2021); Xie et al. (2021); Yuan et al. (2021b); Gu et al. (2022). Moreover, Cheng et al. (2021; 2022) decouple semantic segmentation into mask classification and prediction , which has been widely adopted by state-of-the-art methods in semantic segmentation.

**Plain Vision Transformers for Semantic Segmentation.** Plain ViTs are characterized by a patch embedding layer and stacked transformer layers with a constant sequence length or feature resolution. Consequently, they operate quite differently from conventionally hierarchical architectures. It is worth noting that while several prior works have explored plain ViTs for semantic segmentation Zheng et al. (2021); Ranftl et al. (2021); Strudel et al. (2021); Lin et al. (2022); Chen et al. (2022b); Zhang et al. (2022), this area remains relatively underexplored, particularly when compared to the extensive exploration of semantic segmentation with hierarchical backbones. In recent works, there has been a trend towards increasing complexity in decoders or adapters Lin et al. (2022); Chen et al. (2022b). Simultaneously, in the context of masked image modeling pre-training with plain ViTs Bao et al. (2021); He et al. (2022); Chen et al. (2022a); Peng et al. (2022), there is still a prevalent use of the UperNet decoder for transfer learning in semantic segmentation due to its simplicity. Although SegViT utilizes the off-the-shelf features of plain ViTs to keep the method straight and efficient, it falls short of achieving a similar level of performance as ViT-Adapter-Mask2Former. Furthermore, the dedicated ATM module used in SegViT impacts its generality to some extent. In this study, we develop high-performance 'minimalist' systems to fill in the gaps.

## 3 APPROACH

### 3.1 MOTIVATIONS

Our motivations stem from reconsideration of the existing methods in the context of plain ViTs. As discussed above, the current state-of-the-art ViT-Adapter-MaskFormer is not elegant and efficient for practical semantic segmentation with plain ViTs. It motivates us to develop 'minimalist' systems that incorporate fewer induction biases and feature architectural simplicity. Additionally, we assume that the potent representations learned by MIM plain ViTs will obviate the need for certain sophisticated and dedicated designs. This belief underscores the power of representational learning and inspires us to unearth the underlying principles that render efficacy and promote efficiency in this context.

### 3.2 CASE STUDIES ABOUT UPERNET DECODER

We start with the widely used UperNet decoder to conduct some case studies for more insight. All the experiments are conducted with 80k train iterations and a $512 \times 512$ training crop size on ADE20K. Other settings are identical to those in the original paper of BEiT Bao et al. (2021). As shown in Table 1, the UperNet decoder outperforms the linear decoder by about two points, demonstrating its effectiveness for plain ViTs. The linear decoder incorporates a single linear layer upon the final output of the plain ViT, which is a very simple baseline used in Strudel et al. (2021). We emphasize this as the UperNet decoder has been proven unnecessary for advanced pyramid ViTs Xie et al. (2021); Gu

et al. (2022). Furthermore, we observe that the auxiliary supervision is unnecessary and the usage of the pyramid pooling module (PPM) affects the systematic performance slightly.

After removing the unnecessary components, the UperNet decoder indeed incorporates two main characteristics: utilizing hierarchical features and fusing these features at high resolution. Inspired by recent findings in object detection and human pose estimation that the last feature map of MIM plain ViTs is sufficientLi et al. (2022); Xu et al. (2022), we design a simple up-sampling decoder that only utilizes the last feature map and gradually increases the feature resolution. Specifically, we apply bilinear interpolation twice with $2\times$ up-sampling, followed by a $3\times3$ convolution. Finally, a point-wise convolution is used to obtain the final segmentation map:

$$out = Conv_{1\times1}(Conv_{3\times3}(Up(Conv_{3\times3}(Up(x))))), \qquad (1)$$

where each $Conv_{3\times3}$ is a sequence of Conv, Batch Normalization, and ReLU. We keep the input dimension and output dimension consistent for each $3\times3$ convolution. Table 2 shows that such a simple design outperforms the linear decoder by a large margin. When we reduce the channel number of the UperNet head to bridge the computational gap, the simple decoder performs slightly better. It suggests that up-sampling the last feature map is highly competitive compared to the UperNet decoder and high-resolution features are crucial to high-performance semantic segmentation. Our conclusion can not be drawn from some early works such as SETR where the PUP decoder does not reflect an obvious advantage over the Naive decoder (48.64 $vs.$ 48.18). Given that we utilize plain ViTs with MIM pre-training, it may be caused by different pre-trained weights, different training strategies, or even different details of decoder design.

Table 1: Ablation study on components of the UperNet decoder. 'aux.' denotes the auxiliary supervision of deep features

| decoder | mIoU |
|---|---|
| Linear | 49.6 |
| UperNet | 51.5 |
| UperNet w/o aux. | 51.6 |
| UperNet w/o aux. and PPM | 51.9 |

Table 2: Comparison between the UperNet decoder and our simple up-sampling decoder. 'Slim UperNet' denotes that we reduce the channel number from 768 to 320. The GFLOPs of decoders are reported

| decoder | mIoU | GFLOPs |
|---|---|---|
| Improved UperNet | 51.9 | 493 |
| Simple up-sampling | 51.1 | 110.6 |
| Slim UperNet | 51.0 | 101.4 |

### 3.3 PLAINSEG

In this section, we continue with the idea of simple up-sampling to develop a high-performance 'minimalist' system. As shown in Fig. 1, our approach consists of a pre-trained plain ViT backbone to extract features, a lightweight transformer decoder to classify masks, and a refiner to restore the feature resolution.

**Encoder.** A plain ViT mainly consists of three parts: a patch embedding layer, position embedding, and transformer layers. Among them, position embedding is necessary for the transformer to provide absolute or relative positional information. For instance, MAE He et al. (2022) uses the one-dimensional absolute position embedding while BEiT Bao et al. (2021) adopts the two-dimensional relative position embedding. Two-dimensional position embedding usually performs better in downstream vision tasks so it is widely adopted in object detection and semantic segmentation tasks. Note that it also introduces inductive biases and this is why we emphasize fewer inductive biases rather than none.

**Transformer Decoder.** The motivation for incorporating the transformer decoder into our system comes from the transformer-based encoder-decoder architecture of NLP Vaswani et al. (2017) as well as the more general concept of image segmentation Cheng et al. (2021; 2022). Specifically, our 'minimalist' system is built on the Mask2Former framework Cheng et al. (2022). We observe that the different configurations are employed by ViT-Adapter and SegViT with regard to the width of transformer decoder layers. These two works, to the best of our knowledge, are the only ones that combine plain ViTs with the mask classification paradigm. In detail, ViT-Adapter utilizes a slim transformer decoder with width 256 for ViT-B and a wide transformer decoder with width 1024 for ViT-L while SegViT sets the decoder width to half of the encoder width (384 for ViT-B and

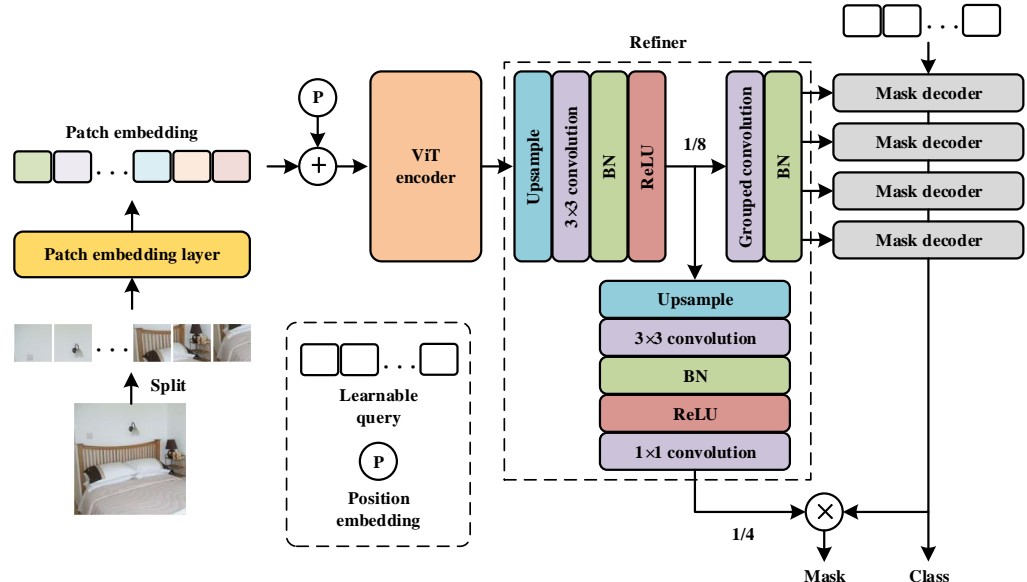

Figure 1: The detailed architecture of PlainSeg.

512 for ViT-L). We note that the width of transformer decoder layers has a substantial impact on parameter efficiency as they are usually stacked many times. Therefore, other than them, we adopt a slim transformer decoder with width 256 for all the models. The original Mask2Former framework heavily relies on the hierarchical features of pyramid backbones so applying it to non-hierarchical vision transformers is not trivial. In the next section, we introduce our refiner which generates high-resolution features and bridges the gap between the transformer encoder and decoder.

**Refiner.** We firstly up-sample the final output of plain ViTs through bilinear interpolation and utilize a 3×3 convolution to refine the feature map:

$$F_{refine} = Act(Norm(Conv_{3 \times 3}(Norm(Up(F_{vit}))))), \tag{2}$$

where $F_{vit}$ denotes the last feature of plain ViT, $Up$ is the bilinear interpolation, $Norm$ is the normalization layer such as BN or LN, and $Act$ is the activation layer. The resulting 1/8 resolution $F_{refine}$ will be used for cross-attention. To align the dimensions, a naive solution is compressing the channels of $F_{vit}$ by the 3×3 convolution. However, it possibly leads to a loss of valid information. Another solution is generating multi-scale feature maps from $F_{vit}$ like ViTDet, and then compressing them to 256 channels. Nevertheless, this will increase architectural complexity and we maintain that lowering feature resolution is unnecessary since we only utilize the last feature map of abundant global semantics. To this end, we propose a simple width-to-depth technique, splitting $F_{refine}$ into several groups along the channel dimension and passing each grouped feature into cross-attention of sequential decoder layers. In detail, we perform a grouped 3×3 convolution based on $F_{refine}$ and split the feature:

$$F_{cross-attn} = Split(Norm(Conv_{3 \times 3, group=n}(F_{refine}))), \tag{3}$$

where $F_{cross-attn}$ is the N-group feature passed into cross-attention and each grouped feature has 256 channels. As shown in Fig. 1, the i-th grouped feature is associated with the i-th transformer decoder layer. This pattern is repeated once in a round robin fashion following Mask2Former. The feature for mask prediction is obtained based on $F_{refine}$ in a similar way:

$$F_{mask} = Conv_{1 \times 1}(Act(Norm(Conv_{3 \times 3}(Up(F_{refine}))))), \tag{4}$$

where the output dimension of $Conv_{3 \times 3}$ is 256 and the last $Conv_{1 \times 1}$ is added following Mask2Former. We use Batch Normalization and ReLU activation following the widely used U-perNet decoder. The group number of $F_{cross-attn}$ is set to 3 for the base model and 4 for the large model. Therefore, the number of transformer decoder layers is 6 for the base model and 8 for the large model.

### 3.4 Improved Learning Rate Strategy

Layer-wise learning rate decay (LLRD) has been widely adopted in fine-tuning MIM plain ViTs. It typically leads to smaller learning rates for shallow layers and larger learning rates for deep layers. In the context of semantic segmentation with plain ViTs, the decoder is usually considered as the 'last layer'. However, it is unreasonable to treat the randomly initialized decoder and the pre-trained transformer layers equally despite the existence of layer-wise decay. Therefore, we introduce a scale factor $s$ greater than 1 for the randomly initialized parameters. For a base learning rate $l$ and decay factor $r$, the learning rates of the decoder and the *i-th* layer of the plain ViT are $l \times s$ and $l \times r^i$ ($i > 1$). We note that employing a larger learning rate for the randomly initialized parameters is a common configuration. However, we highlight these details because they significantly contribute to the effective optimization of slim transformer decoders. To our best knowledge, previous works usually opt to use a larger $l$ along with a smaller $r$ rather than combining a scale factor with LLRD.

### 3.5 PlainSeg-Hier

There are two motivations behind presenting the PlainSeg-Hier. Firstly, spatial details from shallow layers are helpful for fine-grained semantic segmentation. Secondly, we aim to align with SegViT and ViT-Adapter which also utilize hierarchical features and develop a straightforward counterpart. Note that we still adhere to the 'minimalist' pursuit and the practical principles in the development of PlainSeg, including creating high-resolution features by simple up-sampling techniques and using slim transformer decoders. Given the utilization of hierarchical features, we naturally associate it with multi-scale. Specifically, we employ deconvolutions for up-sampling and max pooling for down-sampling, with a reduction in feature map width by half when the size is doubled. To fuse multi-scale features (1/8, 1/16, 1/32) in a minimal yet efficient manner, we adopt a single deformable transformer encoder layer with multi-scale deformable attention Zhu et al. (2021). Finally, we fuse the 1/4 features and the enhanced 1/8 features to obtain the 1/4 mask features and pass the enhanced multi-scale features (1/8, 1/16, 1/32) into the transformer decoder layers. The width of both the deformable transformer encoder and the transformer decoder is set to 256.

Although we have explored the potential of using the last feature map and demonstrated its effectiveness compared to UperNet, we do not claim that hierarchical features are dispensable, as suggested by Li et al. (2022). There are various approaches to fuse hierarchical features and the original FPNLin et al. (2017b) is relatively outdated. In PlainSeg-Hier, we adopt a single deformable transformer encoder layer, which is a simplification of the pixel decoder in Mask2Former. However, it is sufficient to achieve our goals. More advanced designs and techniques may offer further improvements and we leave these for future research endeavors. In essence, our PlainSeg and PlainSeg-Hier serve as powerful baselines for further exploring the necessity of hierarchical features.

## 4 Experiments

### 4.1 Implementation Details

Extensive experiments are conducted on **ADE20K** Zhou et al. (2017), **PASCAL Context** Mottaghi et al. (2014), **COCO-Stuff 10K** Caesar et al. (2018), and **COCO-Stuff 164K** Caesar et al. (2018). We use the *MMSegmentation* Contributors (2020) toolbox for all the experiments. For the selection of pre-trained plain ViTs, we tame the BEiT Bao et al. (2021), BEiTv2 Peng et al. (2022), and EVA-02-L Fang et al. (2023) models. We generally follow the training recipes of their original papers where the UperNet decoder is the default decoder. In the following, we only introduce the differences and more details are in the appendix. Specifically, we multiply by 10 the learning rate of randomly initialized decoder heads and apply gradient clipping following Mask2Former Cheng et al. (2022). In addition, we train the models with fewer iterations to facilitate research (80K, 20K, 20K, 80K iterations for ADE20K, PASCAL Context, COCO-Stuff 10K, COCO-Stuff 164K). The sliding window strategy is adopted for inference and evaluation following previous works. Only single-scale inference results are reported since one of our major considerations is practicality. We conduct all the experiments using eight RTX 3090.

Table 3: Comparisons with ViT-Adapter Chen et al. (2022b) in terms of parameters and test time. We benchmark the test time by the public tool[1] with a single RTX 3090 and batch size 1. ∗ denotes the result is reproduced by ourselves and † denotes the crop size during train and test is 896. In addition to the total parameters (PRM), we report the proportion of randomly initialized parameters to pre-trained parameters (R/P). Note that the test time may not be directly proportional to the test crop size due to the use of sliding window inference

| method | framework | backbone | mIoU(SS) ↑ | PRM (R/P) ↓ | test time ↓ |
|---|---|---|---|---|---|
| ADE20K | | | | | |
| ViT-Adapter∗ | mask cls | BEiT-B | 54.65 | 121M (41%) | 194ms |
| PlainSeg | mask cls | BEiT-B | **55.70** | 105M (22%) | 138ms(**-29**%) |
| PlainSeg-Hier | mask cls | BEiT-B | 54.62 | 106M (23%) | 143ms(**-26**%) |
| ViT-Adapter | mask cls | BEiT-L | 58.32 | 568M (87%) | 535ms |
| PlainSeg | mask cls | BEiT-L | 58.14 | 333M (10%) | 331ms(**-38**%) |
| PlainSeg-Hier | mask cls | BEiT-L | **58.17** | 322M (6%) | 308ms(**-42**%) |
| Pascal Context | | | | | |
| ViT-Adapter | mask cls | BEiT-B | 64.00 | 120M (40%) | 240ms |
| PlainSeg | mask cls | BEiT-B | 63.74 | 105M (22%) | 156ms(**-35**%) |
| PlainSeg-Hier | mask cls | BEiT-B | **64.92** | 105M (22%) | 174ms(**-28**%) |
| ViT-Adapter | mask cls | BEiT-L | 67.79 | 568M (87%) | 578ms |
| PlainSeg | mask cls | BEiT-L | 67.25 | 332M (10%) | 325ms(**-44**%) |
| PlainSeg-Hier | mask cls | BEiT-L | **67.66** | 326M (8%) | 317ms(**-45**%) |
| COCO-Stuff 10K | | | | | |
| ViT-Adapter | mask cls | BEiT-B | 50.00 | 120M (40%) | 149ms |
| PlainSeg | mask cls | BEiT-B | **51.09** | 105M (22%) | 107ms(**-28**%) |
| PlainSeg-Hier | mask cls | BEiT-B | 51.01 | 105M (22%) | 107ms(**-28**%) |
| ViT-Adapter | mask cls | BEiT-L | 53.2 | 568M (87%) | 342ms |
| PlainSeg | mask cls | BEiT-L | **53.02** | 332M (10%) | 195ms(**-43**%) |
| PlainSeg-Hier | mask cls | BEiT-L | 52.99 | 326M (8%) | 188ms(**-45**%) |
| COCO-Stuff 164K | | | | | |
| ViT-Adapter† | mask cls | BEiT-L | 51.68 | 571M (88%) | 1449ms |
| PlainSeg | mask cls | BEiT-L | 51.14 | 333M (10%) | 358ms(**-75**%) |
| PlainSeg-Hier | mask cls | BEiT-L | **51.75** | 327M (8%) | 336ms(**-77**%) |

## 4.2 COMPARISONS WITH STATE-OF-THE-ART METHODS

In this section, we mainly compare our approaches with two state-of-the-art methods, ViT-Adapter-Mask2Former and SegViT. Table 3 shows that our methods achieve highly competitive performance compared to the state-of-the-art ViT-Adapter while significantly reducing the number of parameters and test time. More importantly, our results demonstrate that it is possible to build a 'minimalist' system with comparable performance to complex systems. To better understand the parameter efficiency of our methods, we report the proportion of randomly initialized parameters to pre-trained parameters (R/P). Given that the same backbone is utilized, a lower R/P indicates that our methods have fewer randomly initialized parameters. Remarkably, the R/P of our large models is nearly 10 times lower than that of ViT-L-Adapter-Mask2Former. We note that this reduction in R/P can have a positive impact on label-limited subtasks of semantic segmentation, such as continual semantic segmentation where each new scenario only contains labels for a subset of categories. We further compare PlainSeg-Hier with SegViT which is the previous state-of-the-art method in terms of trade-off between performance and efficiency. As presented in Table 4, our method significantly outperforms SegViT in terms of performance while maintaining decent inference efficiency. Fig. 2 offers a broader perspective by including more models. It can be seen that our methods achieve a new state-of-the-art trade-off, superior to SegViT across different backbones and datasets. Table 5 illustrates the effect of different pre-training strategies. Using plain ViTs with supervised pre-training,

---

[1]https://github.com/open-mmlab/mmsegmentation/blob/main/tools/analysis_tools/benchmark.py

Table 4: Comparisons with SegViT Zhang et al. (2022; 2023) in terms of parameters and test time. We benchmark the test time by the public tool[1] with a single RTX 3090 and batch size 1. ∗ denotes the result is reproduced by our environment with identical training settings

| method | backbone | mIoU | PRM | test time |
|---|---|---|---|---|
| ADE20K | | | | |
| SegViT | BEiTv2-B | 54.0 | 109M | 69ms |
| PlainSeg-Hier | BEiTv2-B | **55.38** | 106M | 99ms |
| SegViT | BEiTv2-L | 58.0 | 345M | 287ms |
| PlainSeg-Hier | BEiTv2-L | **59.77** | 322M | 308ms |
| Pascal Context | | | | |
| SegViT | BEiTv2-L | 66.61 | 344M | 245ms |
| PlainSeg-Hier | BEiTv2-L | **69.60** | 326M | 317ms |
| COCO-Stuff 10K | | | | |
| SegViT | BEiTv2-L | 52.00 | 344M | 158ms |
| PlainSeg-Hier | BEiTv2-L | **54.56** | 326M | 188ms |
| COCO-Stuff 164K | | | | |
| SegViT∗ | BEiT-B | 48.53 | 109M | 120ms |
| PlainSeg | BEiT-B | 49.32 | 105M | 155ms |
| PlainSeg-Hier | BEiT-B | **49.84** | 106M | 155ms |
| SegViT∗ | BEiT-L | 50.17 | 345M | 305ms |
| PlainSeg | BEiT-L | 51.14 | 333M | 358ms |
| PlainSeg-Hier | BEiT-L | **51.75** | 327M | 336ms |

Table 5: Performance with different pre-trained plain ViTs. 'super.' denotes the supervised pre-trained backbones from ViT-AugReg Steiner et al. (2021). We report both single-scale and multi-scale test results for comparison. All the methods employ ViT-L as backbones

| backbone | method | mIoU |
|---|---|---|
| ADE20K | | |
| super. | Segmenter | 51.8/53.6 |
| super. | StructToken | 52.8/54.2 |
| super. | SegViT | 54.6/55.2 |
| super. | ViT-Adapter | 56.8/57.7 |
| super. | PlainSeg | 55.0/56.0 |
| super. | PlainSeg-Hier | 54.7/56.4 |
| EVA-02 | PlainSeg | 61.7/62.0 |
| EVA-02 | PlainSeg-Hier | 61.7/62.0 |
| Pascal Context | | |
| EVA-02 | PlainSeg | 70.2/70.8 |
| EVA-02 | PlainSeg-Hier | 70.6/71.0 |
| COCO-Stuff 164K | | |
| EVA-02 | PlainSeg | 53.4 |
| EVA-02 | PlainSeg-Hier | 53.7 |

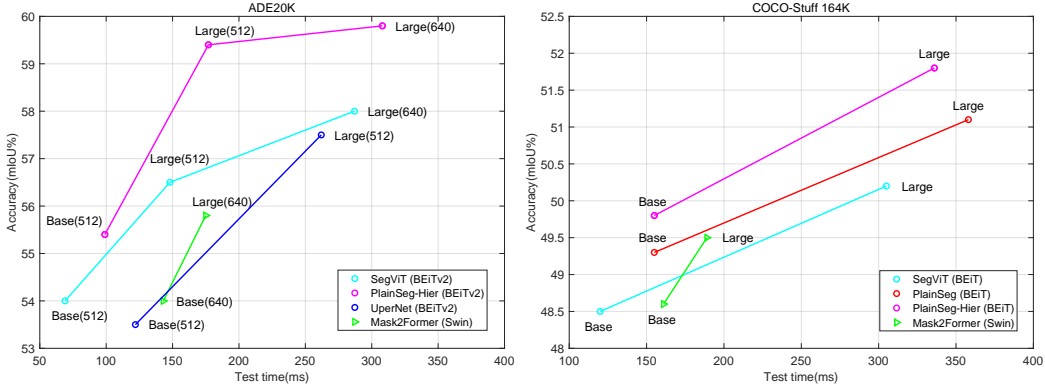

Figure 2: Trade-off between accuracy and test time on ADE20K (left) and COCO-Stuff 164K (right). We exclude methods with significantly lower accuracy. We report the accuracy and test time of sliding window inference for Swin backbones as a reference.

our method outperforms most competitive methods but is inferior to ViT-Adapter. It underscores the effectiveness of our methods and indicates that MIM pre-training narrows and even bridges the gap between our methods and ViT-Adapter-Mask2Former. Furthermore, consistent performance gains on multiple datasets are achieved with EVA-02-L, the current leading open-source plain ViT model utilizing multi-model MIM pre-training on extensive data. It demonstrates the potential of our methods in assessing the transfer ability of potent pre-trained plain ViTs.

Table 6: Ablation study on the refiner. We replace all the 3×3 convolutions with 1×1 convolutions ('w/o 3×3 convolution') or remove all the up-sampling operations ('w/o high resolution') or remove 3×3 convolutions and up-sampling at the same time ('w/o both'). The GFLOPs of decoders are reported

| refiner | mIoU | GFLOPs |
|---|---|---|
| default | 48.7 | 66.0 |
| w/o width-to-depth | 48.4 | 27.3 |
| w/o 3×3 convolution | 48.3 | 14.5 |
| w/o high resolution | 47.8 | 11.9 |
| w/o both | 47.2 | 3.9 |

Table 7: Ablation study on the learning rate and the width of transformer decoder. 'default' denotes we do not apply 10× learning rate to the decoder. 'gradclip' is as same as the gradient clipping in Mask2Former. 'wide decoder' means that the width of transformer decoder is 768 rather than 256. N/A: fail to converge

| Method | mIoU |
|---|---|
| default | 47.5 |
| 4× lr, gradclip | 48.3 |
| 10× lr, gradclip | 48.7 |
| wide decoder | 48.5 |
| wide decoder, 4× lr, gradclip | 48.7 |
| wide decoder, 10× lr, gradclip | N/A |

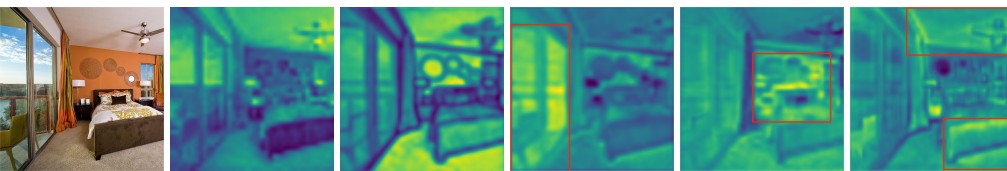

Figure 3: From left to right, they are the input image, visualized feature maps before and after a 3×3 convolution, and visualized feature maps of different groups in Refiner. Different grouped features highlight distinct regions marked with red boxes.

### 4.3 ABLATION STUDY

The ablation study is conducted on COCO-Stuff 164K using BEiT-B backbone. We also train all the models with 80K iterations and a crop size of 512×512. Although PlainSeg is simple in both concept and implementation, we demonstrate that several key architectural designs and training settings contribute to its high performance. In Table 4, we observe a 0.3% decrease in mIoU when directly compressing the output feature of a plain ViT. Either reducing feature resolution or removing 3×3 convolutions has a negative impact on performance and high-resolution features are more crucial. Our approach obtains high-resolution features by directly up-sampling low-resolution features without fusing shallow high-resolution features, showcasing a different paradigm from previous high-performance methods. Table 5 reveals that using a larger learning rate significantly improves the performance of a slim transformer decoder, making it comparable to a wide transformer decoder. However, we do not observe an obvious improvement when applying it to a wide transformer decoder. It can be seen from Fig. 3 that the 3×3 convolution refines the feature maps effectively and the 'width-to-depth' generates more abundant refined feature maps for the transformer decoder. For different groups, the network learns to have a specific focus on different regions.

## 5 CONCLUSION

In this work, we develop high-performance 'minimalist' systems for semantic segmentation with plain ViTs and provide simple and efficient baselines for the field. In the meanwhile, we identify the underlying principles that contribute to the success of our methods. The 'minimalist' systems have many potential benefits. They are more friendly to deployment and acceleration with dedicated devices, facilitating ultimate engineering optimization. They will also help carry out complex multi-model tasks through a single model. We hope our methodology encourages more research toward practical semantic segmentation with plain ViTs. One limitation is that our methods may still be computationally expensive for very high-resolution semantic segmentation tasks. We leave it for future work.

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

# A  APPENDIX

## A.1  IMPLEMENTATION DETAILS

We provide detailed training hyper-parameters in Table 8. We do not search for optimal hyper-parameters on each dataset; instead, we generally follow precedent training recipes and pursue the unified hyper-parameters of different datasets.

Table 8: Training hyper-parameters on four semantic segmentation datasets. Since crop size and total iterations are specific to each dataset, we list every value in the order of ADE20K, PASCAL Context, COCO-Stuff 164K, and COCO-Stuff 10K. Models with EVA-02-L are only trained on the first three datasets

| hyper-parameters | BEiT-B | BEiT-L | BEiTv2-L | EVA-02-L |
|---|---|---|---|---|
| crop size | 640,480,640,512 | | | 640,480,640 |
| learning rate | 3e-5 | 2e-5 | 3e-5 | 2e-5 |
| layer-wise lr decay | 0.90 | 0.95 | 0.90 | 0.90 |
| batch size | 16 | 16 | 16 | 16 |
| total iterations | 80K,20K,80K,20K | | | 80K,20K,120K |
| warm up iterations | 1500 | 1500 | 1500 | 1500 |
| optimizer | AdamW | AdamW | AdamW | AdamW |
| drop path rate | 0.1 | 0.3 | 0.3 | 0.3 |
| weight decay | 0.05 | 0.05 | 0.05 | 0.05 |
| grad clip | 0.01 | 0.01 | 0.01 | 0.01 |

## A.2 MORE DETAILS OF PLAINSEG-HIER

As discussed in Section 3.5, the method to generate multi-scale features is similar to the feature pyramid in BEiT-UperNet Bao et al. (2021) except that we reduce the output width of deconvolutions for higher parameter and computational efficiency. For the features to generate masks, we up-sample the 1/8 output of the deformable transformer encoder by bilinear interpolation and add it to 1/4 features and then use a 3×3 convolution for refinement. In addition, features of three scales are fed into 9 transformer decoder layers in a round robin fashion except for the ViT-L models on ADE20K. We employ 6 transformer decoder layers for them as slight performance drops are observed with more transformer decoder layers.

## A.3 ADDITIONAL EXPERIMENTAL RESULTS

**Detailed Results of Mask2Former (Swin).** Table 9 shows the detailed results of Mask2Former (Swin) on ADE20K and COCO-Stuff 164K.

Table 9: Detailed results of Mask2Former Cheng et al. (2022) on ADE20K and COCO-Stuff 164K. ∗ denotes the result is reproduced by our environment with identical training settings. For ADE20K, we use the models reproduced by *MMSegmentation*. Results in parentheses are obtained with the default whole image inference

| method | backbone | mIoU(SS) | PRM | test time |
|---|---|---|---|---|
| ADE20K | | | | |
| Mask2Former | Swin-B | 54.0(53.9) | 107M | 143ms(94ms) |
| Mask2Former | Swin-L | 55.8(56.1) | 215M | 175ms(115ms) |
| COCO-Stuff 164K | | | | |
| Mask2Former∗ | Swin-B | 48.6 | 107M | 161ms |
| Mask2Former∗ | Swin-L | 49.5 | 216M | 198ms |

**Performance Comparison on Cityscapes.** We conduct experiments on Cityscapes whose lots of thin and small objects pose more challenges to semantic segmentation with plain ViTs. As shown in Table 10, our methods outperform the competitive CNN-based methods, which demonstrates the potential of 'minimalist' systems in challenging traffic scenes. However, they still lag behind advanced segmentation models based on hierarchical ViTs due to the loss of spatial details.

**Comparison with SegViT and UperNet on ADE20K.** We provide detailed results of our methods, SegViT, and UperNet on ADE20K using the same backbone and crop size. As shown in Table

Table 10: Performance comparison on Cityscapes val set. Results in the second line are obtained from *MMSegmentation* model zoo, which are higher than those of original papers in most cases

| Model | mIoU(SS) | Params. |
|-------|----------|---------|
| hierarchical backbones | | |
| FCNLong et al. (2015) | 75.5 | 69M |
| EncNetZhang et al. (2018) | 78.6 | 55M |
| PSPNetZhao et al. (2017) | 79.8 | 68M |
| HRNetWang et al. (2020) | 80.7 | 66M |
| DANetFu et al. (2019) | 80.5 | 69M |
| DeepLabV3+Chen et al. (2018) | 81.0 | 63M |
| OCRNetYuan et al. (2020) | 81.4 | 70M |
| HRFormer-B + OCRYuan et al. (2021b) | 81.9 | 56M |
| SegFormer-B5Xie et al. (2021) | 82.4 | 85M |
| Mask2Former-Swin-BCheng et al. (2022) | 83.3 | 107M |
| Mask2Former-Swin-LCheng et al. (2022) | 83.3 | 215M |
| plain backbones | | |
| PlainSeg-B | 81.7 | 106M |
| PlainSeg-Hier-B | **82.3** | 106M |

11, SegViT only achieves marginal performance improvements upon UperNet while our methods significantly outperform both of them in accuracy.

Table 11: Comparison with SegViT Zhang et al. (2022) and UperNet on ADE20K. ∗ denotes the result is reproduced by our environment with identical training settings

| method | crop size | backbone | mIoU(SS) | PRM | test time |
|--------|-----------|----------|----------|-----|-----------|
| UperNet | 640 | BEiT-B | 53.6 | 163M | 206ms |
| SegViT∗ | 640 | BEiT-B | 53.8 | 109M | 113ms |
| PlainSeg | 640 | BEiT-B | 55.7 | 105M | 138ms |
| PlainSeg-Hier | 640 | BEiT-B | 54.6 | 106M | 143ms |
| UperNet | 512 | BEiTv2-B | 53.5 | 163M | 122ms |
| SegViT | 512 | BEiTv2-B | 54.0 | 109M | 69ms |
| PlainSeg-Hier | 512 | BEiTv2-B | 55.4 | 106M | 99ms |
| UperNet | 512 | BEiTv2-L | 57.5 | 440M | 262ms |
| SegViT | 512 | BEiTv2-L | 56.5 | 344M | 148ms |
| PlainSeg-Hier | 512 | BEiTv2-L | 59.4 | 326M | 177ms |

**Trade-off between Accuracy and Parameters on ADE20K and COCO-Stuff 164K.** As shown in Fig. 4, our methods are superior to SegViT and UperNet.

A.4 VISUALIZATION RESULTS

**Visualized Feature Maps.** Fig. 5 and Fig. 6 show more visualized feature maps about the Refiner.

**Visualized Segmentation Maps.** Fig. 7 and Fig. 8 show the competitive visualization results on ADE20K, PASCAL Context, and Cityscapes. Our methods are competent for segmenting small objects and have decent boundary details.

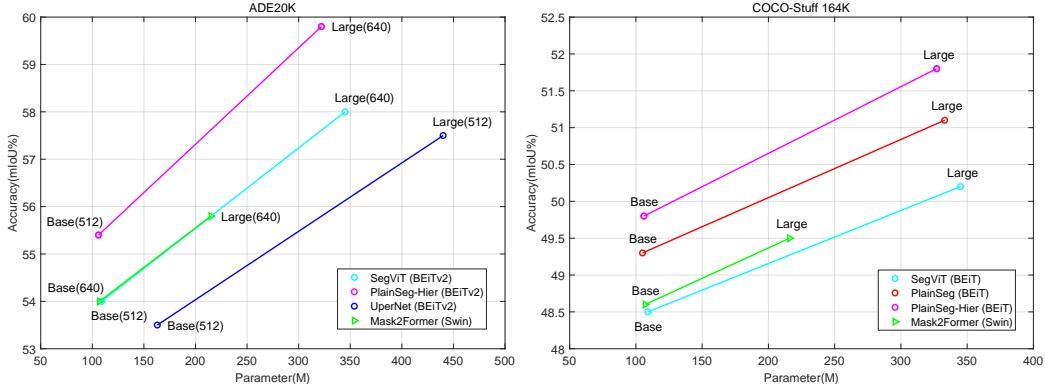

Figure 4: Trade-off between accuracy and parameters on ADE20K (left) and COCO-Stuff 164K (right). We exclude methods with significantly lower accuracy. We report the accuracy of sliding window inference for Swin backbones as a reference.

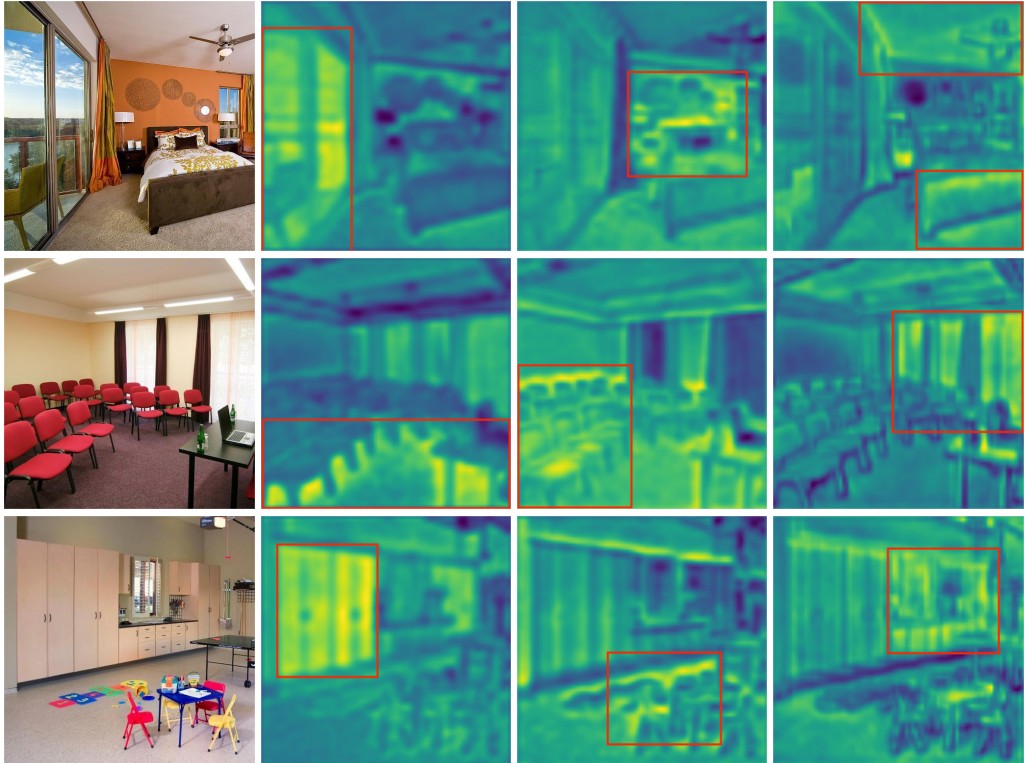

Figure 5: Visualized feature maps of different groups in Refiner. Different grouped features highlight distinct regions marked with red boxes.

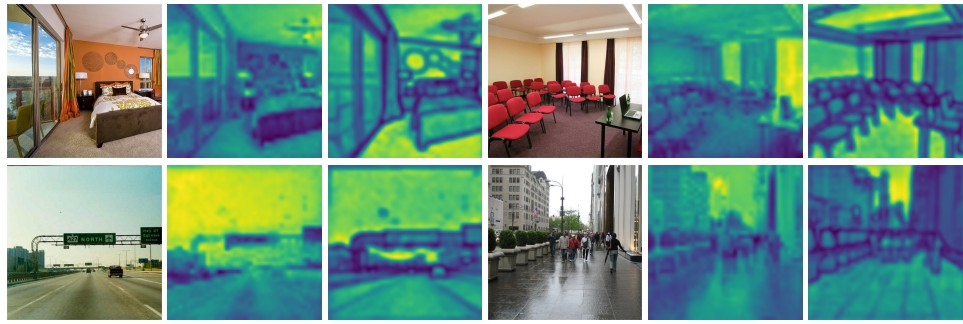

Figure 6: Visualized feature maps before and after a 3×3 convolution.

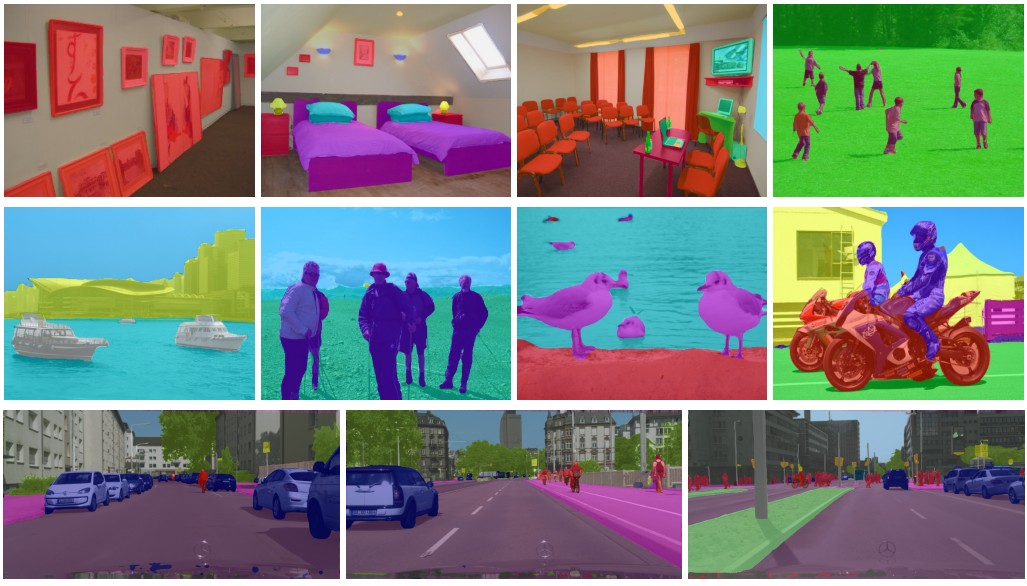

Figure 7: Visualized segmentation maps of PlainSeg on ADE20K (top), PASCAL Context (middle), and Cityscapes (bottom).

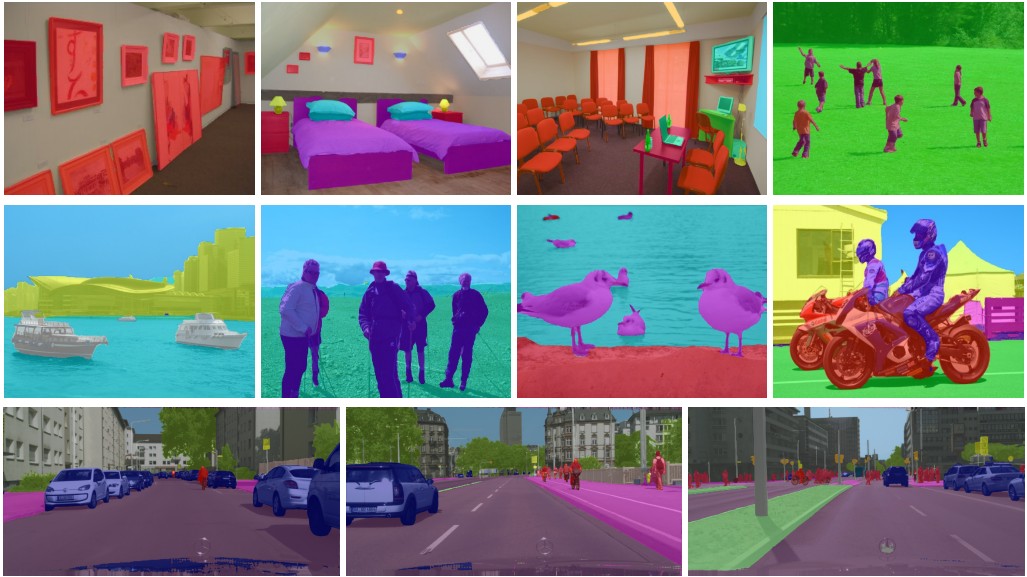

Figure 8: Visualized segmentation maps of PlainSeg-Hier on ADE20K (top), PASCAL Context (middle), and Cityscapes (bottom).

