# OpenReview forum: "Minimalist and High-Performance Semantic Segmentation with Plain Vision Transformers"
_ICLR.cc/2024/Conference — ICLR 2024 Conference Withdrawn Submission_

### Official Review · Reviewer_eJjr · 2023-10-26

**Soundness:** 1 poor
**Presentation:** 2 fair
**Contribution:** 1 poor
**Rating:** 3
**Confidence:** 5

**Summary:**

In this work, plain vision transformers are used to explore a minimalist system for semantic segmentation. Two models are introduced: PlainSeg and PlainSeg-Hier. PlainSeg uses the last feature map only, while PlainSeg-Hier uses the feature hierarchy. In order to achieve a minimalist system, both models use a Mask2Former transformer decoder that omits the pixel decoder portion. Experiments are conducted on four popular datasets.

**Strengths:**

This work's writing is understandable and straightforward.

**Weaknesses:**

The approach makes a relatively small contribution.
- There are two so-called "principles" presented. Nonetheless, the first one—that high-resolution features are essential for semantic segmentation—has received much research and has long been accepted in the field as conventional knowledge. Regarding the second, even in the CNN era, a common training strategy is to use a large learning rate for the newly initialized decoder.
- In terms of the decoder design itself, the core function of the decoder makes no difference from that of Mask2Former. It is in fact a trivial work to adapt an existing decoder design to a shared backbone network, i.e., adapting Mask2Former decoder to plain ViT in this work.
- It makes a difference to divide a shared feature map into several features using group convolution in order to consistently attend to learnable class tokens. However, it is merely an investigation and will not result in a novel contribution.
- There is no substantial contribution from a changing width configuration in the decoder (Sec. 3.3). The deformable transformer layer used in PlainSeg-Hier comes from existing works (Sec. 3.5).
- The experimental experiments are less persuasive since, according to the author, they did not do fair comparisons and did not pinpoint any particular injustices.

**Questions:**

- The reviewer believes that ViT-Adapter employs a bottom-up branch since it begins feature aggregation from the input image and concludes with the final feature from the backbone (line 3, paragraph 1, page 2).
- Typo mistake, which shoube be semantic segmentation (line 2, paragraph 2, page 2).
- It would be preferable to display the linear head FLOPs in Table 2 for comparison.

---

> ### Author Response · Authors · 2023-11-15
> **Response to Reviewer eJjr (1/2)**
>
> Thank you for appreciating our approach. It is very regretful that this work has little value in your opinion. We answer your question one by one and hope that the discussion can help improve our work.
>
> Q1: The approach makes a relatively small contribution.
>
> A1: The purpose of this work is to emphasize the ‘minimalist’ principle and provide simple and efficient baselines for practical semantic segmentation with plain ViTs. We do not aim to introduce a novel technique or approach. We believe our baselines are rewarding for the current semantic segmentation community, serving as a good benchmark for future works and appropriate tools for assessing the transfer ability of base models in semantic segmentation (to replace the commonly used UperNet head).
>
> Q2: There are two so-called "principles" presented. Nonetheless, the first one—that high-resolution features are essential for semantic segmentation—has received much research and has long been accepted in the field as conventional knowledge. Regarding the second, even in the CNN era, a common training strategy is to use a large learning rate for the newly initialized decoder.
>
> A2: For the first one, we aim to emphasize the high-resolution features with simple up-sampling and more importantly without fusing shallow high-resolution features (like the 1/4 or 1/8 feature in the pyramid backbone or the 1/8 feature in the adapter branch of ViT-Adapter). For the second one, although using a large learning rate for the newly initialized decoder is not novel, we have shown that it is useless for a wide transformer decoder adopted by several previous methods.
>
> Q3: In terms of the decoder design itself, the core function of the decoder makes no difference from that of Mask2Former. It is in fact a trivial work to adapt an existing decoder design to a shared backbone network, i.e., adapting Mask2Former decoder to plain ViT in this work.
>
> A3: Since our purpose is to set up a baseline for semantic segmentation with plain ViTs, it is natural to utilize the Mask2Former decoder due to its powerful performance and usage of transformers. We also discuss the performance of different decoders in Section 3.2. On the one hand, we adhere to ‘minimalist’ principles and develop ‘minimalist’ systems. On the other hand, according to the literature, there is still a big gap between semantic segmentation approaches with plain ViTs and those with pyramid ViTs.
>
> Q4: It makes a difference to divide a shared feature map into several features using group convolution in order to consistently attend to learnable class tokens. However, it is merely an investigation and will not result in a novel contribution.
> A4: As stated above, the purpose of this work is to emphasize the ‘minimalist’ principle and provide simple and efficient baselines for practical semantic segmentation with plain ViTs. We tend to introduce simple and effective methods rather than more complex or novel methods.
>
> Q5: There is no substantial contribution from a changing width configuration in the decoder (Sec. 3.3). The deformable transformer layer used in PlainSeg-Hier comes from existing works (Sec. 3.5).
>
> A5: We indeed do not make a substantial technical contribution. However, these changes significantly improve the system efficiency and make our method straightforward.

---

> ### Author Response · Authors · 2023-11-15
> **Response to Reviewer eJjr (2/2)**
>
> Q6: The experimental experiments are less persuasive since, according to the author, they did not do fair comparisons and did not pinpoint any particular injustices.
>
> A6: Maybe we didn't express our ideas accurately in the original manuscript. If one checks out the details of previous works in semantic segmentation with plain ViTs, he will find the fact that there is not a recognized training protocol (especially in terms of learning rates) that is strictly followed by all the approaches. In this work, we conduct numerous experiments on multiple datasets with different backbones. Therefore, it is very difficult to ensure perfectly fair comparisons of all the results. However, we have tried our best to use the common settings and reduce the adjustment of hyper-parameters across different datasets, as presented in Section 4.1 and Table 8. We envision that future works will adopt our training protocol to reduce the training cost significantly.
>
> Q7: The reviewer believes that ViT-Adapter employs a bottom-up branch since it begins feature aggregation from the input image and concludes with the final feature from the backbone (line 3, paragraph 1, page 2).
>
> A7: Thanks for pointing it out. It should be ‘bottom-up’. We mainly aim to emphasize that ViT-Adapter relies on high-resolution features that are directly down-sampled from the input image.
>
> Q8: Typo mistake, which should be be semantic segmentation (line 2, paragraph 2, page 2).
>
> A8: We will examine the manuscript carefully and correct typo mistakes in a subsequent version.
>
> Q9: It would be preferable to display the linear head FLOPs in Table 2 for comparison.
>
> A9: It may not make much sense given the poor performance of the linear head. Anyway, we will add it in a subsequent version.

---

> ### Comment · Reviewer_eJjr · 2023-11-20
>
> Thanks for the response. After carefully reading the manuscript once more and considering the feedback from other reviewers, I still have concerns regarding the technical novelty.
>
> While developing a "minimalist" semantic segmentation system based on plain ViT is a promising beginning (I agree this is a great point), the suggested PlainSeg is less inspiring. (a) The authors appear to cater primarily to the "minimalist" by utilizing the final feature map of the ViT backbone, as stated in the manuscript's main text. However, the resulting PlainSeg remains in the non-simple regime of "backbone + refiner/pixel decoder + mask decoder" and does not constitute a "minimalist" segmentation system. SegViT appears more compact from this angle since it does not include the pixel decoder component. (b) Additionally, PlainSeg-Hier runs counter to your original intention in creating a "minimalist" system only using the final feature map, which makes the whole story less convincing. (c) The majority of the components in the framework are borrowed from Mask2Former and existing FCN-based methods with minor modifications (also suggested by other reviewers). The first emphasized principle has been widely studied and accepted in the community, e.g., FCN and Dilated FCN where the latter leads to a high-resolution feature map for final segmentation.
>
> What exactly constitutes a "minimalist" segmentation system, according to the authors?

---

> ### Author Response · Authors · 2023-11-21
>
> Thank you for the reply. We have nothing to refute your concerns regarding the technical novelty as we do not introduce a novel technique. The responses to your other comments are as follows:
>
> (a): We understand your comparison between PlainSeg and SegViT. Firstly, we recognized the simplicity of SegViT in the original manuscript. The main problem of SegViT is its relatively poor performance. To understand the relationships between PlainSeg and SegViT involves your question at the end. In the context of our work, a “minimalist” segmentation system should incorporate as few inductive biases as possible. Then, it comes to the definition of inductive biases for segmentation. In our opinion, pixel decoder or hierarchical features can be seen as inductive biases. From this perspective, SegViT is also a “minimalist” segmentation system. However, our “minimalist” segmentation system is also high-performance. Other than SegViT, we emphasize high-resolution features more than hierarchical features. It can be seen that PlainSeg with the final feature map has outperformed SegViT. In addition, we would like to explain that our pixel decoder has been very simple and its primary purpose is to form high-resolution features.
>
> (b): Compared to PlainSeg, PlainSeg-Hier does make some compromises. However, it is very important to note that the “minimalist” or the “simplicity” is relative and relies on context. Strictly speaking, our “minimalist” systems are relative to the current state-of-the-art system, namely ViT-Adapter-Mask2Former. If you compare the early LeNet, all the current methods are too complicated. Although PlainSeg-Hier is a by-product of PlainSeg, we show it given that our primary purpose is to provide simple and efficient baselines for practical semantic segmentation with plain ViTs.
>
> (c): It seems that only one reviewer argues that our method is very similar to FCN and we have provided the detailed response. FCN is a milestone and a far-reaching work. However, we emphasize the high-resolution features up-sampled from a single feature map like ViTDet.

---

### Official Review · Reviewer_a5pX · 2023-10-31

**Soundness:** 2 fair
**Presentation:** 2 fair
**Contribution:** 2 fair
**Rating:** 3
**Confidence:** 4

**Summary:**

This paper introduces PlainSeg, a minimalist semantic segmentation system that employs plain Vision Transformers (ViTs) in lieu of more complex, state-of-the-art models. Unlike previous art SegViT, which uses Mask-to-Attention (ATM), PlainSeg incorporates a sequence of three 3x3 convolutional layers. The system demonstrates its efficacy on multiple benchmarks, including the Pascal-context, ADE20K, and COCO-Stuff datasets. The paper argues two main principles: (1) high-resolution features are essential for performance, even when simple up-sampling techniques are used, and (2) a slim transformer decoder demands a significantly higher learning rate compared to a wide transformer decoder. The authors also introduce an extension, PlainSeg-Hier, designed to utilize hierarchical features. Overall, the proposed approach not only achieves competitive performance but also serves as an efficient baseline for semantic segmentation

The proposed approach achieved the best results in the Pascal-context dataset, and it has shown strong performance over its own selected baselines. The datasets are ADE20K and COCO-Stuff 10K/164K.

**Strengths:**

The proposed PlainSeg model excels in its simplicity and reproducibility, offering a straightforward path for implementation. It has demonstrated robust performance across multiple datasets, including Pascal Context, ADE20K, and COCO-Stuff, thereby validating its practical utility. Key advantages include:
1) Outperforming SegViT in both efficiency and effectiveness, and
2) serving as a strong benchmark that can guide future work in semantic segmentation.

**Weaknesses:**

1. The paper has some shortcomings with respect to its claims of novelty. Specifically, the approach appears to be a minor modification of SegViT and borrows heavily from the foundational Fully Convolutional Networks (FCN). The performance gains over SegViT are also not markedly significant, raising questions about the impact of these changes.

2. Additionally, the paper's presentation could be improved; it rigidly adheres to a 3x3 convolution followed by Batch Normalization and a 1x1 convolution, a setup akin to that used in FCN. The lack of exploration beyond these hard-coded parameters suggests that the work might be better framed as an experimental report rather than a ICLR paper.

3. While the paper presents extensive experimental results to emphasize the advantages of its simple up-sampling decoder, the approach lacks novelty. Fully Convolutional Networks (FCN) have already established the use of simple up-sampling as a widely recognized baseline in convolutional neural network-based semantic segmentation.

**Questions:**

1. The paper seems to borrow heavily from Fully Convolutional Networks (FCN), particularly in its use of up-sampling techniques. However, FCN's contributions are not adequately acknowledged or discussed. Could you elaborate on why FCN, a foundational work in semantic segmentation with different backbones like VGG and ResNet, wasn't given its due credit, especially when the proposed approach seems to be an adaptation of FCN with a ViT backbone. Any explanation on this?

2. The paper primarily focuses on datasets that don't require fine-grained semantic segmentation, making me skeptical about the model's capability to handle intricate object boundaries. It would be interesting to have the authors to test the proposed approach on datasets requiring more detailed semantic segmentation, e.g. Pascal VOC? How does the proposed work against the other works with denseCRF or diluted Convolution?

3. A more technical question: when using bilinear interpolation twice in the proposed methods, could authors specify the exact implementation and parameters used? Was the bilinear interpolation learned or pre-defined? Did you use PyTorch's native bilinear interpolation, or some other implementation? Different interpolation techniques can produce different results, it would be interesting to see if this makes a difference on the final results.

---

> ### Author Response · Authors · 2023-11-15
> **Response to Reviewer a5pX (1/2)**
>
> Thank you for appreciating our approach. We answer your question one by one.
>
> Q1: The paper has some shortcomings with respect to its claims of novelty.
>
> A1: The purpose of this work is to emphasize the ‘minimalist’ principle and provide simple and efficient baselines for practical semantic segmentation with plain ViTs. We do not aim to introduce a novel technique or approach. We believe our baselines are rewarding for the current semantic segmentation community, serving as a good benchmark for future works and appropriate tools for assessing the transfer ability of base models in semantic segmentation (to replace the commonly used UperNet head).
>
> Q2: Specifically, the approach appears to be a minor modification of SegViT and borrows heavily from the foundational Fully Convolutional Networks (FCN). The performance gains over SegViT are also not markedly significant, raising questions about the impact of these changes.
>
> A2: The main contribution of SegViT is the Mask-to-Attention module which is not adopted by our method. We believe that the dedicated design of the transform decoder for semantic segmentation is unnecessary. In addition, we emphasize the significance of high-resolution features which is overlooked in SegViT. The performance gains over SegViT can be clearly observed in Table 4 and Figure 2. As to FCN, the only thing in common may be the usage of 3x3 convolutions which are presented in almost all computer vision papers. We elaborate on the detailed differences in the following answers.
>
> Q3: Additionally, the paper's presentation could be improved; it rigidly adheres to a 3x3 convolution followed by Batch Normalization and a 1x1 convolution, a setup akin to that used in FCN. The lack of exploration beyond these hard-coded parameters suggests that the work might be better framed as an experimental report rather than a ICLR paper.
>
> A3: We employ the common 3x3 convolution precisely because of its simplicity. As answered above, the purpose of this work is to emphasize the ‘minimalist’ principle and provide simple and efficient baselines for practical semantic segmentation with plain ViTs. In fact, we do not emphasize the advantage of 3x3 convolutions except for the simplicity and the 3x3 convolution is not the critical factor contributing to the success of our method. It can be replaced by any more advanced operators. However, we find that the performance gain is very limited for PlainSeg.
>
> Q4: While the paper presents extensive experimental results to emphasize the advantages of its simple up-sampling decoder, the approach lacks novelty. Fully Convolutional Networks (FCN) have already established the use of simple up-sampling as a widely recognized baseline in convolutional neural network-based semantic segmentation.
>
> A4: We want to clarify that the substantial difference between PlainSeg and FCN is whether to fuse shallow high-resolution features. By the ‘simple up-sampling’, we aim to emphasize that we directly up-sample the deepest feature to predict segmentation maps without fusing shallow high-resolution features. In other words, removing the skip connections in FCN will definitely lead to serious performance degradation. From a bigger perspective, our PlainSeg is a non-hierarchical method and removes the usage of hierarchical features that are widely adopted by previous methods including the FCN.
>
> Q5: The paper seems to borrow heavily from Fully Convolutional Networks (FCN), particularly in its use of up-sampling techniques. However, FCN's contributions are not adequately acknowledged or discussed. Could you elaborate on why FCN, a foundational work in semantic segmentation with different backbones like VGG and ResNet, wasn't given its due credit, especially when the proposed approach seems to be an adaptation of FCN with a ViT backbone. Any explanation on this?
>
> A5: We have clarified the substantial difference between our method and FCN. And we approve that FCN is a pioneering work at the beginning of the paper. In addition, the plain ViT backbone indeed matters because it is essentially different from the conventional hierarchical backbones such as VGG and ResNet. We sincerely hope that you can consider related works in semantic segmentation with plain ViTs when evaluating the contribution of the proposed method.

---

> ### Author Response · Authors · 2023-11-15
> **Response to Reviewer a5pX (2/2)**
>
> Q6: The paper primarily focuses on datasets that don't require fine-grained semantic segmentation, making me skeptical about the model's capability to handle intricate object boundaries. It would be interesting to have the authors to test the proposed approach on datasets requiring more detailed semantic segmentation, e.g. Pascal VOC? How does the proposed work against the other works with denseCRF or diluted Convolution?
>
> A6: Pascal VOC is rarely used in recent works. We conduct additional experiments on Cityscapes to evaluate the performance of fine-grained semantic segmentation. These results can be found in Table 10 of the Appendix. Our methods are also competitive. The compared method DeepLabV3+ utilizes dilated convolutions.
>
> Q7: A more technical question: when using bilinear interpolation twice in the proposed methods, could authors specify the exact implementation and parameters used? Was the bilinear interpolation learned or pre-defined? Did you use PyTorch's native bilinear interpolation, or some other implementation? Different interpolation techniques can produce different results, it would be interesting to see if this makes a difference on the final results.
>
> A7: We use PyTorch's native bilinear interpolation for simplicity.

---

### Official Review · Reviewer_Vw3B · 2023-11-01

**Soundness:** 3 good
**Presentation:** 1 poor
**Contribution:** 2 fair
**Rating:** 5
**Confidence:** 4

**Summary:**

The paper proposes a plain ViT-based semantic segmentation approach. Unlike many other state-of-the-art approaches that create some form of hierarchical ViT to utilize the hierarchical features for the segmentation task, the proposed PlainSeg approach is building a minimalistic and simple decoder on top of a plain ViT model. In essence the paper explores how to create a lightweight variant of the mask2former module that can be attached to a ViT, without the need for hierarchical features. The resulting model performs competitively while at the same time not requiring any architectural modifications of the underlying ViT. In addition to that, a slightly more complicated, but stronger PlainSeg-Hier is proposed, that does create hierarchical features from the ViT to feed it into the attached segmentation decoder.

**Strengths:**

- I like the main goal of the paper, to utilize a ViT and put a simple and minimalistic network on top of it in order to do semantic segmentation. The paper shows that this is possible while remaining competitive and knowing that a simpler model is an option is always good for future research.
- In direct comparison to ViT-Adapter, the proposed method seems to perform on par, while requiring fewer additional parameters in the decoder and being computationally more efficient. In comparison with SegViT the model performs better, albeit it is a bit slower.
- The ablations regarding the learning rate reveal some interesting effects.

**Weaknesses:**

- What is the main take-away message and contribution? As I said before, I'm happy to know a simple decoder can be put on top of a plain ViT, but how simple is this decoder really? The initial decoding of feature maps with convs is of course trivial, but what follows seems to be very similar to a full-fledged Mask2Former with minor simplifications. All of a sudden, it doesn't feel so simple anymore. How was this model trained? Using the standard Mask2Former recipe, including Hungarian matching and masked cross-attention? Even if we disregard how simple or complex the decoder is, it's still a vanilla ViT and that is interesting, but this is also where I feel that the novelty ends. It would have been more valuable if we would gain some insights why exactly this architecture is good or should be chosen, but the paper here just shows that with slight modifications an existing decoder can be put on top of a plain ViT. As such it feels a bit more like an engineering paper and there is little real novel insights to be gained.

- Partially the paper is hard to follow and some stuff is simply not described very well, but it's rather assumed the reader just knows. For example:
  - The short section on PlainSeg-Hier is very confusing and to me this architecture still isn't completely clear. Even with the additional explanation in the appendix, I can't say I for sure I now know how this network looks like.
  - Sometimes there are missing citations, for example if you write that "Layer-wise learning rate decay (LLRD) has been widely adopted", please give some citations for that. Which "sliding window strategy" do you use? Or when you write "On this basis, we further present PlainSeg-Hier which incorporates hierarchical features, as seen in previous works." which works do you mean?
  - At some point an ablation writes about "width-to-depth", while it's not clear what this really means
  - As stated above, how closely is the training process aligned to that of Mask2Former and where does it diverge?
  - What are the feature map visualizations? Magnitudes of the features? PCA projections? Random channels?

- For some of the tables I'm unclear where the baseline numbers come from. For example for Table 3 I can't find the respective numbers in the ViT-Adapater paper. Even for settings directly reported in the paper, e.g. ADE20k results, I can't find a setup with 568M parameters getting a 58.32 mIoU, so how were these numbers obtained? And how representative are they for the original approaches?

- In a certain way, the Segment Anything Model (SAM) also uses a simple ViT and a super small decoder on top. I know that SAM is promptable and it's not really great at spitting out semantic segmentations, nevertheless the core idea is rather similar and it should be discussed in related work.

**Questions:**

- Large part of the paper is about PlainSeg, but the short section on PlainSeg-Hier introduces another variant that is mostly used as the main contender in the tables. What is the real value of PlainSeg and why not discuss PlainSeg-Hier in more detail?

- Why the strong focus on MIM? Couldn't other pretraining methods such as DINO(v2) work too?

- You write that you use deconvolutions. I thought by now this term has vanished and nobody uses it anymore. Please use transposed convolutions instead, since deconvolutions are something completely different!

- Equation 2 has an additional Norm that is missing from Figure 1. Which of these is correct?

- There are quite some weird terms and typos, e.g. "to render efficacy", although maybe my English just isn't that great, but also "objective detection", "induction bias" and "multi-model data". Maybe run this through a grammar/spell checker.

---

> ### Author Response · Authors · 2023-11-15
> **Response to Reviewer Vw3B (1/2)**
>
> Thank you for appreciating our approach and the affirmation of our main goal. We answer your question one by one.
>
> Q1: What is the main take-away message and contribution?
>
> A1: The purpose of this work is to emphasize the ‘minimalist’ principle and provide simple and efficient baselines for practical semantic segmentation with plain ViTs.
>
> Q2: The usage of Mask2Former decoder.
>
> A2: The Mask2Former decoder indeed contains two parts: the pixel decoder and the transformer decoder. Our simplification focuses on the pixel decoder rather than the transformer decoder. Simplicity is always relative and in the context of our work, it is equal to the minimization of inductive biases. We incorporate the transformer decoder because it introduces little inductive biases and retains versatility. Compared to the current state-of-the-art system ViT-Adapter, our methods are still surprisingly simple. By the way, the original transformer[1] in NLP contains both encoder layers and decoder layers.
>
> Q3: It would have been more valuable if we would gain some insights why exactly this architecture is good or should be chosen.
>
> A3: We present two principles in the Abstract, which are two underlying factors behind the success of our methods. Hope this will help you better understand our methods. The reasons for choosing our architecture are the ‘minimalist’ (minimization of inductive biases, simple architectures for deployment) and the high-performance (the best trade-off between accuracy and inference time).
>
> Q4: The short section on PlainSeg-Hier is very confusing and to me this architecture still isn't completely clear. Even with the additional explanation in the appendix, I can't say I for sure I now know how this network looks like.
>
> A4: We will provide a more detailed introduction of PlainSeg-Hier in a subsequent version. Could you tell us a certain confusion for a better clarification?
>
> Q5: Sometimes there are missing citations, for example if you write that "Layer-wise learning rate decay (LLRD) has been widely adopted", please give some citations for that. Which "sliding window strategy" do you use? Or when you write "On this basis, we further present PlainSeg-Hier which incorporates hierarchical features, as seen in previous works." which works do you mean?
>
> A5: Thank you for pointing out these problems. We will provide detailed citations in a subsequent version.
>
> Q6: At some point an ablation writes about "width-to-depth", while it's not clear what this really means.
>
> A6: The ‘width-to-depth’ is firstly used in the introduction of the refiner, Section 3.3.
>
> Q7: As stated above, how closely is the training process aligned to that of Mask2Former and where does it diverge?
>
> A7: As answered above, we retain the transformer decoder of Mask2Former. Therefore, the overall training process is similar to that of Mask2Former.
>
> Q8: What are the feature map visualizations? Magnitudes of the features? PCA projections? Random channels?
>
> A8: We provide the visualized feature maps in Figure 3.
>
> Q9: For some of the tables I'm unclear where the baseline numbers come from. For example for Table 3 I can't find the respective numbers in the ViT-Adapater paper. Even for settings directly reported in the paper, e.g. ADE20k results, I can't find a setup with 568M parameters getting a 58.32 mIoU, so how were these numbers obtained? And how representative are they for the original approaches?
>
> A9: It seems that the authors of ViT-Adapter removed this setup in the accepted version of ICLR 2023 as they also highlighted the performance on detection tasks. Nonetheless, all the results (except for the results reproduced by ourselves) can be obtained from the Appendix of the original arxiv version (https://arxiv.org/pdf/2205.08534v1.pdf) or the official github repository (https://github.com/czczup/ViT-Adapter/tree/main/segmentation). We can ensure that these results are representative of ViT-Adapter on semantic segmentation tasks.
>
> Q10: In a certain way, the Segment Anything Model (SAM) also uses a simple ViT and a super small decoder on top. I know that SAM is promptable and it's not really great at spitting out semantic segmentations, nevertheless the core idea is rather similar and it should be discussed in related work.
>
> A10: We will add the SAM in related work.
>
> [1]Vaswani A, Shazeer N, Parmar N, et al. Attention is all you need[J]. Advances in neural information processing systems, 2017, 30.

---

> > ### Comment · Reviewer_Vw3B · 2023-11-15
> >
> > A2: I think it's rather weird to compare a vanilla transformer decoder layer in an NLP Transformer with the transformer decoder of a Mask2Former. For sure the underlying code might be very similar, but the detailed workings and what they learn are very different. Here the transformer decoder of the Mask2Former still feels pretty complex and heavy compared to some simple convolution-based method. (But yes, simplicity does rely on context.)
> >
> > A4: Upon reading the two sections again I know actually feel I would likely be able to guess how you mean it, but either just add a simple diagram, or describe it in equations like Eq. 2/3/4. You explain the ops that you use and roughly where you use them, but if you are more precise it will be a lot easier to follow.
> >
> > A13: We'll sadly this still seems to be more present than expected. ViTDet also simply used the wrong name. If you check the ViTDet code, you can see that their "deconvolution" is actually a transposed convolution. I'm just pointing out that you should be using the correct term, even if some other people falsely call it deconvolution too.

---

> > > ### Author Response · Authors · 2023-11-16
> > >
> > > Thank you for the quick reply.
> > >
> > > A2: There are indeed some differences between Mask2Former transformer decoders and transformer decoders in NLP. However, the main difference comes from the training phase and Mask2Former just uses the standard pytorch implementation of transformer decoders. Another reason we incorporate the transformer decoder of Mask2Former is that decoupling classification and mask prediction seems to be becoming a trend. As you can see, two main methods of comparison (SegViT and ViT-Adapter) also benefit from it. We believe that it can reflects the timeliness of our benchmark.
> > >
> > > A4: We have understood your demand and will improve the representation of PlainSeg-Hier in a subsequent version.
> > >
> > > A13: You are right that we have no reason to follow the wrong expression if we know it is wrong. We will correct it and thank you for pointing it out.

---

> ### Author Response · Authors · 2023-11-15
> **Response to Reviewer Vw3B (2/2)**
>
> Q11: Large part of the paper is about PlainSeg, but the short section on PlainSeg-Hier introduces another variant that is mostly used as the main contender in the tables. What is the real value of PlainSeg and why not discuss PlainSeg-Hier in more detail?
>
> A11: We report the results of PlainSeg and PlainSeg-Hier in Table 3 and Table 5. For Table 4, we only report the results of PlainSeg-Hier to make a fair comparison with SegViT as it also uses hierarchical features. PlainSeg and PlainSeg-Hier follow the same design principle and both of them are representative instances of ‘minimalist’ and high-performance systems. The only difference is whether to use the hierarchical features. We primarily discuss the PlainSeg because it is more consistent with the ‘minimalist’ principle. The main purpose of PlainSeg-Hier is to benchmark  hierarchical methods and make a fair comparison with SegViT.
>
> Q12: Why the strong focus on MIM? Couldn't other pretraining methods such as DINO(v2) work too?
>
> A12: Theoretically, our method can work with any pre-trained plain ViTs. We focus on MIM just because it is of great influence and a diverse range of MIM plain ViT models have been pre-trained with extensive datasets. DINO is OK but DINOv2 may be not appropriate. Because our method fine-tunes the backbones while DINOv2 emphasizes the performance of frozen backbones.
>
> Q13: You write that you use deconvolutions. I thought by now this term has vanished and nobody uses it anymore. Please use transposed convolutions instead, since deconvolutions are something completely different!
>
> A13: We use deconvolutions following the ViTDet[1]. We will correct this in a subsequent version.
>
> Q14: Equation 2 has an additional Norm that is missing from Figure 1. Which of these is correct?
>
> A14: Thanks for the careful reading and sorry for the issue. The Equation is right and we will correct the figure.
>
> Q15: There are quite some weird terms and typos, e.g. "to render efficacy", although maybe my English just isn't that great, but also "objective detection", "induction bias" and "multi-model data". Maybe run this through a grammar/spell checker.
>
> A15: Sorry for the typos and thank you for pointing them out. We are sure to correct them in a subsequent version.
>
> [1]Li Y, Mao H, Girshick R, et al. Exploring plain vision transformer backbones for object detection[C]//European Conference on Computer Vision. Cham: Springer Nature Switzerland, 2022: 280-296.

---

### Official Review · Reviewer_iRsh · 2023-11-10

**Soundness:** 2 fair
**Presentation:** 3 good
**Contribution:** 2 fair
**Rating:** 3
**Confidence:** 5

**Summary:**

This paper focuses on the problem of semantic image segmentation. The author proposes a high-performance ‘minimalist’  segmentation system based on plain ViTs. Meanwhile, the paper also offers some insights into the practical principles for adapting plain ViTs to semantic segmentation task. The experimental evaluations are conducted on several popular semantic image segmentation datasets and the experimental results look good. It achieves highly competitive performance compared to ViT-Adapter and outperforms SegViT.

**Strengths:**

1. The experimental evaluations are conducted on four popular and challenging semantic image segmentation datasets ADE20K, Pascal Context, COCO-Stuff 10K, and COCO-Stuff 164K, and the experimental results (accuracy, parameters, efficiency) seem good when compared to previous high-performance solutions.
2. Overall, the writing and illustrations are clear and easy to follow.

**Weaknesses:**

1. The major concern of the whole paper is its technical novelty. The author claims two findings and principles: (1) high-resolution features are important for high performance (2) the slim transformer decoder requires a much larger learning rate than the wide one. While the first point is obvious and has already been mentioned by various previous algorithms, and the second point could be a training trick, rather than something inspirational. Combining plain ViTs with a hierarchical structure is not new, hierarchical design already exists in many other frameworks, and the whole pipeline in Figure 1 is not inspiring and appealing, and is hard to motivate potential readers.
2. Some of the content and paper organization need to be adjusted. For example, in the introduction section, the former two paragraphs are more like to be a paragraph in the related work section. While it emphasizes more on the motivation of the work. In Table 3, some lower performance results are highlighted, it is something stranger and needs to be modified.

**Questions:**

In-depth analysis and inspirational designs are needed to make the whole paper and algorithm more appealing, and thus can motivate potential readers.

---

> ### Author Response · Authors · 2023-11-15
> **Response to Reviewer iRsh**
>
> Thank you for appreciating our approach. We answer your question one by one.
>
> Q1: The major concern of the whole paper is its technical novelty.
>
> A1: The purpose of this work is to emphasize the ‘minimalist’ principle and provide simple and efficient baselines for practical semantic segmentation with plain ViTs. We do not aim to introduce a novel technique. We believe our baselines are rewarding for the current semantic segmentation community, serving as a good benchmark for future works and appropriate tools for assessing the transfer ability of base models in semantic segmentation (to replace the commonly used UperNet head).
>
> Q2: The author claims two findings and principles: (1) high-resolution features are important for high performance (2) the slim transformer decoder requires a much larger learning rate than the wide one. While the first point is obvious and has already been mentioned by various previous algorithms, and the second point could be a training trick, rather than something inspirational.
>
> A2: We believe that we still provide some extra insights given that our method is not exactly the same as the previous works you mentioned. For the first point, our high-resolution features are obtained by simple up-sampling of deep features without fusing shallow high-resolution features, which should be different from most previous works. As you can see, ViT-Adapter utilized an additional branch to extract high-resolution features while SegViT only leveraged the 1/16 features. Intuitively, simply up-sampling low-resolution features can not restore the spatial details for semantic segmentation. It used to be considered unable to achieve high performance. In our work, we empirically demonstrate that such a method is able to be competitive and report the best performance of non-hierarchical methods. For the second point, our finding is overlooked by previous works. It could be more inspirational if we provide a deeper analysis of this phenomenon. However, this is out of the scope of the work given our original purpose.
>
>
> Q3: Combining plain ViTs with a hierarchical structure is not new, hierarchical design already exists in many other frameworks, and the whole pipeline in Figure 1 is not inspiring and appealing, and is hard to motivate potential readers.
>
> We do not claim the novelty of hierarchical design in our paper. 	We benchmark both non-hierarchical and hierarchical semantic segmentation with plain ViTs. The whole pipeline in Figure 1 showcases a ‘minimalist’ architecture for high-performance semantic segmentation. The ‘minimalist’ principle is the core of the original ViT[3] which has been a milestone work with great influence.
>
> Q4: Some of the content and paper organization need to be adjusted. For example, in the introduction section, the former two paragraphs are more like to be a paragraph in the related work section. While it emphasizes more on the motivation of the work. In Table 3, some lower performance results are highlighted, it is something stranger and needs to be modified.
>
> Thank you for pointing out the problems. We will correct them in a subsequent version.